# Cohort study of a specialist social worker intervention on hospital use for patients at risk of long stay

Sonya Osborne,[1] Gai Harrison,[2] Angela O'Malia,[2] Adrian Gerard Barnett,[1] Hannah E Carter,[1] Nicholas Graves[1]

[1]Australian Centre for Health Services Innovation, Institute of Health and Biomedical Innovation, Queensland University of Technology, Kelvin Grove, Queensland, Australia
[2]Department of Social Work, Royal Brisbane and Women's Hospital, Herston, Queensland, Australia

**Correspondence to**
Hannah E Carter;
HANNAH.CARTER@QUT.EDU.AU

## ABSTRACT

**Background** Long-stay patients in acute hospitals commonly present with complex psychosocial needs and use high levels of hospital resources.

**Objective** To determine whether a specialist social worker-led model of care was associated with a reduction in length of stay for medically stable patients with complex psychosocial needs who were at risk of long stay, and to determine the economic value of this model relative to the decision makers' willingness to pay for bed days released.

**Design** A prospective, matched cohort study with historical controls.

**Setting** A large, tertiary teaching and referral hospital in metropolitan Southeast Queensland, Australia.

**Methods** Length of hospital stay for a cohort of patients seen under the specialist social worker-led model of care was compared with a matched control group of patients admitted to the hospital prior to the introduction of the new model of care using a multistate model with the social worker model of care as an intermediate event. Costs associated with the model of care were calculated and an estimate of the 'cost per bed day' was produced.

**Results** The model of care reduced mean length of stay by 33 days. This translated to 9999 bed days released over 12 months. The cost to achieve this was estimated to be $A229 000 over 12 months. The cost per bed day released was $23, which is below estimates of hospital decision makers' willingness to pay for a bed day to be released for an alternate use.

**Conclusions** The specialist social worker-led model of care was associated with a reduced length of stay at a relatively low cost. This is likely to represent a cost-effective use of hospital resources. The limitations of our historic control cohort selection mean that results should be interpreted with caution. Further research is needed to confirm these findings.

## Strengths and limitations of this study

► This is the first study to evaluate a hospital-based social work initiative from an economic perspective.
► The small sample size contributed to relatively high levels of uncertainty in the results.
► The use of historical controls who were selected on a different set of criteria to the intervention cohort may have introduced bias or confounding effects beyond those that were controlled for.
► Additional benefits of the model of care, including the potential for improved patient outcomes as well as a reduced workload for existing social workers, were not included in our analysis.

## INTRODUCTION

The longer patients remain in hospital after they are medically ready for discharge, the worse the situation becomes for the patient, their families and for the health system. Patients classified as 'long stay' in acute hospitals are at risk of poor health outcomes and use up valuable hospital resources.[1–3] A retrospective observational study of over 22 000 patients concluded that patients staying in hospital 14 days or longer suffered increased in-hospital morbidity and mortality.[4] Deconditioning and loss of mobility are common outcomes for inpatients who are medically fit for discharge but remain in hospital, with older people being particularly at risk.[5] For patients aged 65 or older, length of stay increased the chance of developing geriatric syndromes in hospital, such as pressure ulcers, incontinence, falls, functional decline and delirium; independent of physical, cognitive or functional impairment.[5] In addition, older people's mobility, muscle strength and aerobic capacity can be adversely affected by just 10 days of bed rest, which, alarmingly, translates into almost 10 years of functional decline.[6] In recognition of these poor health outcomes, hospitals strive to ensure that patients are discharged in a timely manner once they are deemed medically stable, or fit for discharge, that is, once the medical decision has been made that the patient has completed the required acute care, including all relevant investigations with none further anticipated, and is ready to be discharged from the hospital.

Long stay, that is, the prolonged hospitalisation of patients who are medically fit for

discharge, is often the result of non-medical or complex psychosocial barriers to discharge. Patients at risk of long stay commonly present with complex psychosocial needs.[7] Some long-stay patients remain in hospital because they lack family support, have limited access to community services or are waiting to be placed in an interim or long-term care facility.[8–10] Making the unanticipated transition from hospital to institutional care represents a significant life event for many patients and their families, with associated financial and personal costs that may delay discharge. Patients assessed as no longer having capacity to make decisions about their care may similarly face an extended period of hospitalisation, especially if a guardian needs to be appointed to manage their affairs.[11] Material and social disadvantages such as homelessness are also barriers to discharge, while organisational inefficiencies and lack of residential care options may result in placement delays.[12 13] Without timely intervention, patients who present with complex social needs and vulnerabilities are at risk of being stranded in hospital.[13]

Typically, long-stay patients are referred to hospital social workers, who are spending increasing amounts of time working with this patient cohort to address discharge barriers.[9 14–16] However, there is limited evidence on the economic value of hospital social work services, and even less evidence on the economic value of social work initiatives targeting long-stay patients.

The purpose of this paper is to report on an evaluation of an innovative specialist social worker-led model of care designed to identify and coordinate timely discharge of long-stay patients and those at risk of a long stay in a large teaching hospital. We estimated the cost of delivering the model of care and reported its benefits in terms of bed days released. An assessment of the economic value of the intervention was made using previously reported estimates of Australian hospital decision makers' maximum willingness to pay for a bed day to be released for an alternate use. We used the Consolidated Health Economic Evaluation Reporting Standards to guide our study report.[17]

## METHODS
### Establishing a profile of long-stay patients
Prior to developing any intervention to address the issue of long-stay patients, it was critical to establish a profile of this vulnerable patient cohort. In March 2016, a snapshot 1-day audit of several hospital administrative databases was conducted to identify acute patients whose length of stay was equal to or greater than 21 days and subacute patients whose length of stay was equal to or greater than 35 days. Patients in intensive care units, maternity wards, mental health wards and the emergency department (ED) were excluded from the audit. A total of 93 patients met the inclusion criteria. Using a purpose-designed audit tool, two persons, a nurse navigator and a senior social worker, independently reviewed each chart to identify barriers to discharge. Eleven psychosocial

barriers were identified as risk factors for long stay. These risk factors included family complexity, dysfunction or conflict; domestic and family violence; need for substitute decision making; carer stress; substance abuse; homelessness; challenging patient behaviours; disability; patient or family financial stress; need for residential aged care placement and disability care planning. These risk factors were in turn able to inform the early identification of patients at risk of long stay. Further explanations of the psychosocial barriers to discharge and several theoretical case study exemplars of the psychosocial complexities of long-stay patients can be found in online supplementary files 1 and 2.

### Study design, setting and participants
A prospective cohort study with historical controls design was conducted in a large 929 bed, tertiary teaching and referral hospital in metropolitan Southeast Queensland, Australia to evaluate a specialist social worker-led model of care. The primary objective was to determine the association of the specialist social worker-led model of care with lengths of stay for patients who were medically stable and fit for discharge before and after introduction of the new model of care. The secondary objective was to determine the economic value of the model of care by estimating the cost per bed day released. We took the perspective of the hospital decision maker and their willingness to pay for a bed day released for an alternate use.

The patient cohort included all patients who were entered onto a purpose-built clinical case management database called Pathfinder and managed by the specialist social worker over a 3-month period from June to August 2016. A historical control group was assembled from hospital administrative databases by selecting a cohort of patients who received usual care before the specialist social worker-led model of care commenced. In order to select a similar group of patients, control patients were those in acute or subacute wards who met the definition of a long-stay patient and who had received a social work intervention.

### Usual model of care (control)
Prior to the introduction of the new model of care, long-stay patients were case managed solely by the allocated ward based social worker who typically held up to 40 cases at any one time of both acute short-term and long-stay patients. The social work intervention for all cases included psychosocial assessment with view to managing appropriate and timely discharge. Typically referrals for long-stay patients were made to social work once the patient was medically stable. There were no early screening processes to identify long-stay patients so early intervention was not undertaken. Acute short-term patients were often prioritised to maintain rapid patient flow yet there was no framework in place to manage the multiple tasks, meetings, stakeholder liaison, reports and paperwork associated with the care of long-stay patients. The service was reactive rather than proactive, there was

no visibility over delays to social work intervention. Importantly, there was no formal escalation process in place to resolve systemic issues so individual social workers typically attempted to resolve these matters independently, often without success, resulting in lengthy delays.

### Specialist social worker-led (new) model of care (intervention)

In addition to the individual case management by ward social workers, an additional level of specialised service delivery was added to enhance individual social worker support and expedite discharge through a multifaceted approach. A process was introduced whereby ward social workers could refer patients they believed to be at risk of long stay to the specialist social worker via clinical consultation. These referrals were subsequently entered on an electronic patient tracking system called Pathfinder by the specialist social worker. This allowed the senior social worker to monitor patients at risk of prolonged hospitalisation. The tracking programme was a core component of the model of care and was updated daily by ward social workers. As a clinical case management tool, it recorded psychosocial and systemic barriers to discharge as well as current length of stay, which allowed continuous monitoring of patient outcomes. The specialist social worker also regularly collated a list of long-stay patients from hospital databases and then consulted with the different social work teams to identify those patients facing psychosocial barriers to discharge. In contrast to the old model of care, the new model was proactive rather than reactive and adopted a systematic approach to identifying and monitoring long-stay patients or those at risk of long stay. Key components of the specialist service are listed below, with further detail in online supplementary file 3:

1. A highly trained and skilled specialist social worker to undertake an overarching advisory role.
2. A social worker assistant to support the specialist social worker, particularly to manage administrative tasks.
3. The development of a clinical case management and reporting database called 'Pathfinder' to allow the specialist social worker to design tailored interventions for each patient based on their unique barriers to discharge, and to monitor, track, manage and oversee all patients on the residential aged care, adults with disability and medico-legal pathways.
4. Change in communication and reporting processes (eg, ward social nurses provided regular progress updates to the social worker assistant who recorded the information into Pathfinder; and the specialist social worker regularly communicated through daily reports to hospital executive and other major stakeholders).
5. Development of clear internal escalation processes for 'stranded patients' to engage hospital executives in making timely and appropriate decisions about patient movement and transfer.
6. Implementation of pathways and partnerships with key agencies and care providers to ensure timely and appropriate discharge, including the development of a nursing home vacancy register.

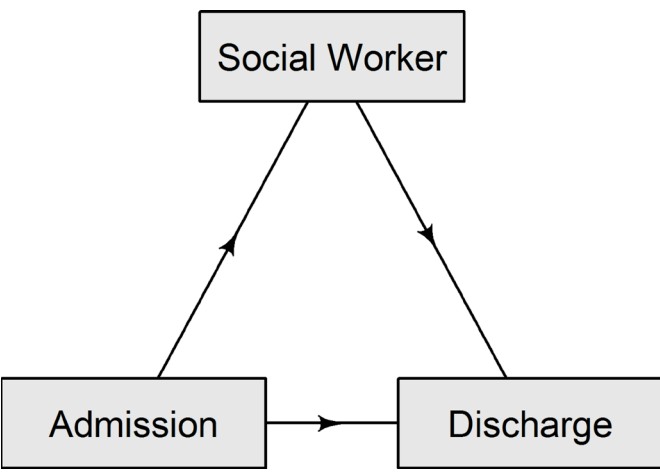

**Figure 1** The three states used to measure the change in length of stay due to the social worker intervention.

7. More robust data collection on key performance indicators.

### Statistical and economic analysis

Characteristics of the specialist social worker-led model of care cohort and the control group were compared using cross-tabulations and summary statistics. Differences between the two groups were tested using the t-test for continuous variables like age, and a $\chi^2$ test for categorical variables like gender. All analyses were made using the R software package (V.3.1.0). The difference in length of stay between the patients treated under the new model of care and the control group of patients was estimated using a multistate model.

To estimate the reduction in length of stay we used the models developed by Beyersmann *et al*[18] with the new model of care as an intermediate state between admission and discharge (see figure 1). Confidence intervals for the length of stay were generated by bootstrapping the data and recalculating the difference in length of stay associated with the social worker intervention. We used 1000 bootstrap estimates.

The bootstrap length of stay estimates were applied in a Monte Carlo simulation to model the cost and length of stay outcomes over a 12-month period. The costs associated with the new model of care are reported in 2017 Australian dollars and include one full-time equivalent specialist social worker for 12 months at a total employment cost of $158 000 plus one full time equivalent social work assistant for 12 months at a total employment cost of $71 000. These costs were assumed to be known with certainty. Discounting of costs was not necessary as the reported time horizon was only 12 months. The activity of the team was reported to be in a range of 20–30 patients managed per month, and to model this we used a uniform distribution between 20 and 30. One thousand simulations were made from all distributions to account for the uncertainty in the results. An estimate of the 'cost per bed day' was produced to inform judgements around the value for money of the new model of care. This metric

**Table 1** Participant demographics

| | Usual model of care | Specialist social worker-led model of care |
|---|---|---|
| | (n=60) | (n=52) |
| Age, mean (SD) | 69 (18) | 74 (14) |
| Gender, n (%) | | |
| Female | 23 (38) | 23 (44) |
| Male | 37 (62) | 29 (56) |

recognises that the value of a bed day released may differ according to the decision maker.[19] We used estimates from a recent survey of Australian hospital chief executive officers which revealed they are willing to pay an average of $216 for a ward bed day released from a quality improving activity.[19]

## Patient and public involvement

The study did not involve contact with individual patients. Patients were not involved in setting the research question or the outcome measures, nor were they involved in the design and implementation of the study. Given the vulnerability of the patient cohort we do not plan to disseminate results directly to patients.

## RESULTS

A total of 112 patients were included; 52 patients in the specialist social worker-led model of care group and 60 patients in the control group. As presented in table 1, there were some differences between the two groups. The new model of care group was older, with mean difference of 5 years (95% CI −1 to 11 years, p value 0.11). The ratio of men to women was similar in the two groups ($\chi^2$ test p value 0.66).

The average length of stay for the entire sample was 70 days and ranged from 3 to 422 days. The median time between admission and involvement of the social worker was 25 days, with an interquartile range from 9 to 50 days.

The mean number of bed days expected to be released per patient managed under the new model of care was 33. The distribution around this result was skewed and ranged from 5 to 70 over 1000 Monte Carlo simulations (figure 2).

The cost to run the model of care for 12 months was $228 876. Based on the modelled simulations, a total of 9999 bed days were released over 12 months at an average cost of $23 per bed day released (table 2).

When applying the finding that decision makers are willing to pay $216 to release a bed day there was a 100% probability (1000/1000 simulations) that this model of care will meet the decision criterion. There was an 82% probability (820/1000 simulations) that the cost of saving 1 bed day was less than $100.

## DISCUSSION

This is the first study to evaluate a hospital-based social work initiative from an economic perspective. We found a specialist social worker-led model of care reduced the

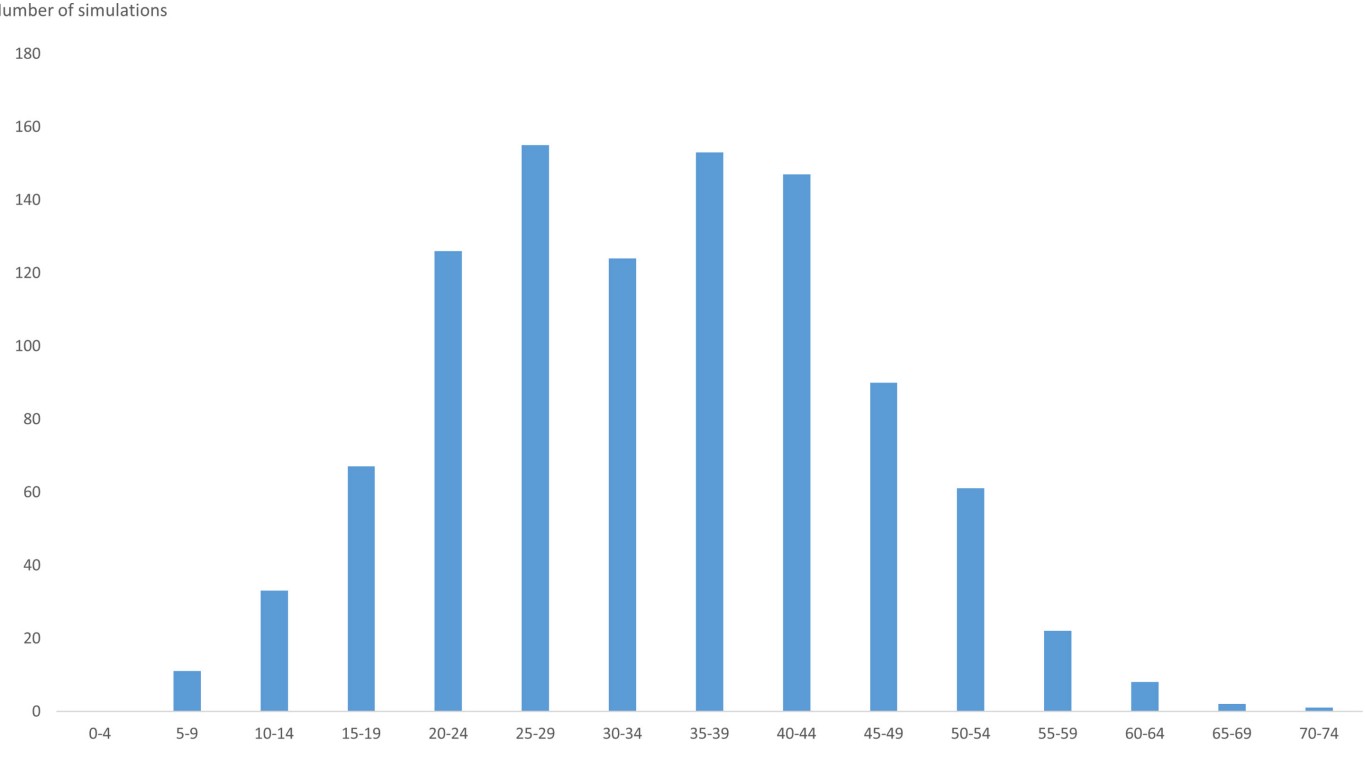

**Figure 2** Histogram of the bed days released per patient due to the social worker intervention.

| Table 2 | Results of 1000 Monte Carlo simulations from the input parameter distributions | | | |
|---|---|---|---|---|
| | Mean | SD | Min | Max |
| Total bed days released per patient managed | 33 | 12 | 5 | 70 |
| Average number of patients managed per month | 25 | 3 | 20 | 30 |
| Total bed days released per year | 9999 | 3781 | 1140 | 24 408 |
| Average cost to save 1 bed day | $23 | $16 | $9 | $201 |

average length of stay by 33 days, with 9999 bed days released over 12 months. This translated to a cost of $23 per bed day released. By definition, this study cannot be considered a cost-effectiveness analysis as we did not explicitly account for the value of patients' health outcomes. However, length of stay is now a well-recognised measure of resource use and efficiency in hospitals, with delayed discharge resulting in reduced patient flow and 'access block' for new admissions.[20] Applying the findings of Page et al[19] that hospital decision makers are willing to pay up to $216 to release 1 bed day, our results indicate there was a 100% chance that adopting the specialist social worker-led model of care described in this study was the right decision, and by extension likely to represent a cost-effective use of resources. This result is supported by the reality that the study hospital has now formally adopted and funded this model of care on a permanent basis.

The new model of care prioritised early identification of patients at risk of long stay, patient tracking, proactive intervention and system level responses. These four factors promoted a more coordinated and targeted social work response to addressing psychosocial barriers to discharge which in turn assisted in expediting hospital discharge. In addition, early identification of potential long-stay patients was facilitated by clinicians' greater awareness of the main psychosocial barriers to discharge when conducting initial psychosocial assessments.

Although the tendency is for hospital social workers to be referred the most complex patients requiring time-intensive interventions,[15] hospital-based social workers are under increased pressure to demonstrate that the services they deliver are both effective and cost-efficient.[9 14 15 21] Notably, there has been very limited research conducted on the economic value of social work in hospitals. Auerbach et al described the 'cost-containment' value of social work, which they defined as how effective social workers are 'in both serving patients and keeping hospital costs down'.[14] A 2001 study reported on a modelled cost–benefit analysis of social work services in the ED setting, and found that dedicated social work staffing of EDs may yield net economic benefits to a hospital system, especially in large urban centres.[22] However, a literature review did not identify any economic evaluations of hospital-based social

work initiatives. This may be a reflection of the interpersonal nature of the profession in which outcomes have traditionally been measured subjectively or qualitatively. However, as hospital decision making becomes increasingly focused on the provision of high value care,[23 24] it is important for the social work discipline to demonstrate the economic, as well as clinical, value of their work.

The comparability of intervention and control groups is a key weakness of our study. Control patients were selected after having met the definition of long stay, while intervention patients were able to be selected without first meeting this definition. This approach enabled us to best reflect the change in practice that occurred given the retrospective nature of our analysis. Specifically, there was no process for the early identification of patients at risk of long stay prior to the intervention, with ward social workers typically becoming involved once patients had met the definition of long stay and were experiencing barriers to discharge. Following the introduction of the new model of care, a focus on early identification meant that patients could be identified, tracked and monitored prior to them meeting the long-stay definition. We acknowledge that these differences may have introduced selection bias in the length of stay results. A prospective pre–post study design with clear criteria about eligible patients and a staff member employed to monitor patients and systematically determine their inclusion would have allowed for the selection of more comparable controls. The sample size of both cohorts was also relatively small which contributed to the uncertainty in cost per bed day results. Not included in this analysis was the potential for the specialist social worker-led model of care to lead to improved patient clinical and quality of life outcomes, as well as a reduced workload for the existing social workers employed by the hospital.

The study was undertaken at a large metropolitan hospital where an oversight role was warranted by the volume of patients being seen. It is unknown whether the efficiencies generated by this role would be replicable in smaller hospital settings. However, the model is scalable with a number of discrete elements that can be adopted separately. There has been interest from other health services in this new model and to date some elements of the model have been adopted by two regional hospitals and one interstate urban hospital.

In conclusion, we found that a specialist social worker-led model of care reduced length of stay in an acute hospital setting at a cost that has been reported as acceptable to hospital decision makers. The model of care is therefore likely to represent a cost-effective use of hospital resources.

**Acknowledgements** We acknowledge Sarah Napier, who was instrumental in developing the specialist social worker-led model of care for long-stay patients at study hospital, along with Laura Farrelly and Helen Jones who assisted with data collection.

**Contributors** GH and AO'M made substantial contributions to conception and design of the study. AGB, HEC and SO provided critical review and contributed feedback on the final protocol and ethics application. AGB conceived the statistical

analysis plan and conducted statistical analysis and interpretation of data. HEC conceived the economic analysis plan. NG contributed to the economic data analysis plan and conducted the evaluation and interpretation of data. SO and HEC drafted the final manuscript and critically revised for important intellectual content. All authors read, revised and approved the final manuscript.

**Funding** Funding in the form of in-kind support was provided under a partnership agreement between the study hospital and the Australian Centre for Health Services Innovation (AusHSI).

**Competing interests** None declared.

**Patient consent for publication** Not required.

**Ethics approval** The study was approved by the relevant Human Research Ethics Committees for the hospital (HREC/16/QRBW/476) and the university (QUT1600001102).

**Provenance and peer review** Not commissioned; externally peer reviewed.

**Data sharing statement** The full dataset used, which has no identifying information, is available from the corresponding author.

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
