## [Reviewer comments · BMJ Open]

ARTICLE DETAILS

TITLE (PROVISIONAL)	A cohort study of a specialist social worker intervention on hospital utilisation for patients at risk of long stay
AUTHORS	Osborne, Sonya; Harrison, Gai; O'Malia, Angela; Barnett, Adrian; Carter, Hannah; Graves, Nicholas

VERSION 1 – REVIEW

REVIEWER	Peter May Trinity College Dublin, Ireland
REVIEW RETURNED	12-Apr-2018

GENERAL COMMENTS	This article examines the impact of a new social worker model on hospital utilisation for “medically stable patients with complex psychosocial needs who were at risk for long stay.” Its key strength is being among the first studies to evaluate a social worker intervention impact on utilisation, and the results suggest scope for significant savings and improvement to patient experience for a priority population. However, there are a number of issues requiring attention or clarification before the article merits publication. MAJOR COMMENTS 1 Eligibility criteria More detail and clarity are required on the participants in both the treatment group and the historical comparison group. The abstract describes the population as “medically stable patients with complex psychosocial needs who were at risk for long stay”. However, the phrase “medically stable” is neither defined nor used at any point past the introduction. And “at risk of long stay” is also unclear. At the critical point for defining eligibility (Methods>Participants on page 5), you write “The patient cohort included all patients who were entered onto a purpose-built clinical case management database called Pathfinder and managed by the specialist social worker over a three month period from...”. On what basis were patients put onto Pathfinder? And what caused them to be managed by the specialist social worker? Who were they; in particular how were “medical stability” and “risk of long-term stay” defined? Then the comparison group participants are described as “meeting pre-identified criteria determining their risk of long stay”. What were the “long stay” criteria for comparators? What do you know about their “medical stability”? 2 Eligibility, outcome and bias
---

Following on from the previous point, how did the treatment group score on “pre-identified criteria determining their risk of long stay”? This is critical to the integrity of the study when LOS is the outcome of interest – if comparison group patients were identified on the basis of factors that predict LOS, and the treatment group differs systematically to the comparison group on these factors, then observed differences in outcome are more or less inevitable. For both groups, clarify whether eligibility variables are baseline (hospital admission), pre-intervention (occur between admission and intervention) or outcome (e.g. LOS). Defining the sample by outcomes such as LOS is problematic in terms of bias and policy relevance (<https://www.ncbi.nlm.nih.gov/pmc/articles/PMC5034210/>). Consider presenting both groups according to all “pre-determined criteria” of long-stay risk in Table 1.

3 Timing bias

The paper is strengthened by interest in timing-related bias but in reading the Wolkewitz reference I struggled to follow how your study is relevant to “length bias”. This seems to be a particular concern when there is a “Time-dependent entry criterion”, i.e. an additional event has to occur between enrolment and exposure, e.g. being put on a ventilator where ventilator-related infection is the exposure of interest (or Oscar nomination in the Oscar example).

I don’t understand what your time-dependent entry criterion is, unless it’s some kind of flag “this person is now at risk of long stay” that indicates social workers need but does not automatically trigger social worker involvement; if this is the case it is not sufficiently explained. If adjusting for “length bias”, consider an additional supplementary table where you follow Wolkewitz et al’s schematic Table 2 (page 1177) and explain for each field how your study corresponds to the Oscar example.

You don’t mention Wolkewitz’s other concern “time-dependent exposure bias” but look again at this. As written your paper gives no indication of difference in time-to-contact with social worker across the two groups. But if this differs there are a number of interpretations. On the one hand, if time-to-intervention is different between groups then this risks other biases whereby one group were in hospital longer (and so nearer to discharge, *ceteris paribus*) irrespective of the intervention. On the other hand, intervention timing has been shown to be key to impact of consultative hospital interventions on utilisation (<https://www.ncbi.nlm.nih.gov/pmc/articles/PMC5034210/>), and if the time-to-contact is less in the treatment group this could itself be an indicator of intervention efficiency (assuming the right people are being identified).

Consider presenting both groups’ time-to-contact with social worker in Table 1: “The timing of all events should be recorded. Careful analysis and reporting of important details of the statistical analysis (including time origin and entry time) are required in the presence of time dependent study entries and exposures.” [What is new? Page 1172)

4 Cohort matching and treatment effect estimation

Concerns about eligibility criteria are compounded by large differences in characteristics of the treatment and comparison groups. At a bare minimum this requires more discussion in the limitations – do the authors think this variation occurred by chance or reflects systematic differences?

As well as the big age difference, the comparison group come from internal medicine and surgery wards, but treatment group came almost exclusively from internal medicine wards (Table 1). As well as an obvious collinearity risk, this begs the question: if your treatment group are (almost) all from the internal medicine wards, why not make internal medicine ward location an eligibility criterion for the comparison group? You'd ditch unrepresentative controls while substantively preserving your treatment group. The status quo invites heterogeneity between groups that is hard to justify (or, at least, is not justified currently).

Also you seem to have included 'missing ward data' as a predictor in regression (Table 2). This requires more justification, especially as it's only for one subject. Are you controlling for some unobserved factor associated with this missingness? What happens if you drop this?

Both age and ward have a significant association with LOS in the primary analysis (Table 2). This seems a significant concern worth flagging in the limitations given that the groups differ on these factors. While these covariates were controlled for in multivariate regression the authors should consider using a matching technique such as propensity score weights to minimise confounding worries and isolate the estimated effect of the intervention (J Pain Symptom Manage. 2014 Oct;48(4):711-8).

5 Statistical analysis

Primary analysis is summarised page 7, lines 43-52. Not clear why a survival analysis is used for a utilization variable. Utilization data are statistically awkward but log-transformation is not a particularly good way to address this (J Health Econ. 1998 Jun;17(3):283-95). Why not use nonlinear models, bootstrap the standard errors and report the estimated effect in days?

(https://www.york.ac.uk/media/economics/documents/herc/wp/10_01.pdf)

I am also confused by the use of percentage reductions instead of day reductions. This appears a way to overcome the retransformation problem (J Health Econ. 1998 Jun;17(3):283-95) but I don't see that it does. A percentage is a ratio, but the ratio of $\ln(x)$ to $\ln(y)$ is not the same as the ratio of x to y so what is the advantage of using %? In what sense are % differences calculated on the log scale relevant?

If % reductions are retained, justify this choice in more detail with respect to the retransformation problem and clarify in reporting if these are % of total LOS or % of LOS from the point of social worker interaction per "length bias" adjustment.

Secondary analysis (later presented in Table 3) is not adequately covered pages 7-8. Explain both the aim and the approach in more detail; currently "Monte Carlo" appears for the first time in Results. See also next comment.

6 Table 3

I got lost around Table 3. The results are summarised as "The cost to run the model of care for 12 months was \$228,876. Based on the modelled simulations, a total of 7,165 bed days were released over 12 months at an average cost of \$78 per bed day released (Table 3)."

My reading of this is that if 7,165 bed days were saved at a cost of \$228,876 then the cost to release a bed day is $(228876/7165) = \$32$. You need to spell out how you reached the \$78 figure because it is not intuitive from the results given. Per previous comment, better explanation of methods might help here.

	Also, how are estimated bed days calculated? You explicitly stay away from days in the primary analysis (Table 2), reporting instead % reductions. If calculating days from log-transformed output is a problem in Table 2, how is it not in Table 3? The introduction of the \$216 here is in any case too abrupt and inappropriate. If comparing cost per bed day saved against this \$216 benchmark was an aim at the outset, mention this in both the Introduction and the Methods. If it occurred only ex post then leave it for the Discussion and delete lines 13-21 on page 9. MINOR COMMENTS 1 'Economic evaluation' The title of this article identifies it as an economic evaluation and this phrase recurs in framing the work conducted. Economic evaluation typically involves the assessment of a treatment's impact on costs (and often but not necessarily outcomes). The primary aim of this study is assessing effect on hospital LOS, and then this is reported in % terms. The authors concede this is not a CEA, since no outcomes are included, but an economic evaluation without outcomes is a *cost*-minimisation analysis. This primary analysis includes no costs. The authors add that "length of stay is now a well-recognised measure of resource use and efficiency in hospitals" (page 10, line 33). This may justify the study question but it does not make it an economic evaluation. Since LOS already appears in the title, consider changing "economic evaluation" to "cohort study" and removing all mention of economic evaluation from the text. 2 Social worker involvement The manuscript provides good detail on the social worker model once patients are under social worker auspices and the supplementary file expands upon this. But I didn't understand how the social worker comes to be involved in treating a patient. Are they identifying suitable patients unilaterally via hospital databases and approaching teams with primary responsibility for patient care? Or are they involved at the invitation of that team? Or something else? And did this differ in the new model to the old? This is particularly important given that being seen by the social worker seems a prerequisite for membership of treatment and comparison groups (MAJOR COMMENTS 1 & 2). 3 Who are the population to target? Fundamentally, the purpose of a study such as this is to answer the question 'should we give intervention I to population P?' The inadequate description of the population is a concern not only for rigour (MAJOR COMMENTS 1 & 2) but also usefulness to decision-makers. By the end of the study, the suggestion is that this model of hospital social work should be pursued but we have little information on who are the people this model will help. Only hospitals with such patients should employ this model, presumably, so we need to know more about who they are and how we identify them. 4 How does it work? Some interpretation would also be welcome. The discussion focuses on contextual justification and resource concerns, which are all valid. But why do the authors think they see the association that they do? Consider adding some information on how better social work provision expedites hospital discharge for this particular population (while avoiding causal language). Also, consider
--	--

	commenting on how to identify these “at risk of long stay” patients to minimise their unnecessary stays. This material could replace the paragraph beginning line 46 page 9, which essentially restates why conducting the study is important; that paragraph could be merged into the introduction or excluded. 5 Miscellanea -Time period is described as three months May to August (line 24, page 6). Give specific dates or this is a four-month period. -Sample size calculation is described as “86% power to detect a 40% decrease in length of stay assuming a two-sided 5% type 1 error” (page 6, line 34). If you are powering for detecting change only in a specific direction, isn’t this a one-sided test? -Recruited sample (n=53) is mentioned in Methods (line 44, page 7). This should not be revealed until the results. -Sensitivity analyses. Were there any? Usually a good idea in utilization studies, especially with small samples. For example, given that the treatment group came almost exclusively from general medicine wards (Table 1), what happens if surgery patients are dropped from analysis?
--	---

REVIEWER	Riyaz Jinnah MD FRCS Wake Forest school of medicine, North Carolina, USA
REVIEW RETURNED	10-May-2018

GENERAL COMMENTS	This is an excellent topic for research. My main objection is with the methodology. The authors have used a historical control and they discuss this. They have 60 patients of which 59 are medical and 1 surgical vs all medical in the study group. It would strengthen the study significantly if only medical patients were used and the stats redone.
--

VERSION 1 – AUTHOR RESPONSE

Response to Reviewers’ Comments

Reviewer 1 comments	Response	Changes to manuscript
MAJOR COMMENTS		
1 Eligibility criteria.		
More detail and clarity are required on the participants in both the treatment group and the historical comparison group. The abstract describes the population as “medically stable patients with complex psychosocial needs who were at risk for long stay”. However, the phrase “medically stable” is neither defined nor used at any point past the introduction. And “at risk of long stay” is also unclear.	Response: We thank the reviewer for their feedback and have added further detail within the text of the document defining ‘medically stable’, and consequently and consistently referring to these patients as medically fit for discharge. In addition, long stay’ has also been defined. The terms ‘acute’ and ‘subacute have also been defined in footnote 1 and 2. Finally, we have provided further detail of the risk factor criteria for at risk for long stay; that is the 11 psychosocial barriers to	Changes: Under introduction, para 1, last sentence Under introduction, para 2, line 1

	discharge identified in the audit to determine a profile of long stay patients.	
At the critical point for defining eligibility (Methods>Participants on page 5), you write “The patient cohort included all patients who were entered onto a purpose-built clinical case management database called Pathfinder and managed by the specialist social worker over a three month period from....”. On what basis were patients put onto Pathfinder? And what caused them to be managed by the specialist social worker? Who were they; in particular how were “medical stability” and “risk of long-term stay” defined?	Response: We thank the reviewers for their feedback and have added further detail in the text by adding a section titled, ‘Establishing a Profile of Long Stay Patients’ . We think this paragraph addresses the specific reviewer queries regarding how patients came to be identified as being at risk of long stay. Further elaboration on how patients were put into Pathfinder was made in the section Specialist Social-worker led model of care	Changes: Under Methods, newly added para 1 footnotes 1 and 2 Under Specialist Social Worker Model of Care, para 1 In addition, we have added 2 additional supplementary files (3 in total)
Then the comparison group participants are described as “meeting pre-identified criteria determining their risk of long stay”. What were the “long stay” criteria for comparators? What do you know about their “medical stability”?	Response: The statement that comparison group participants met pre-identified criteria determining their risk of long stay was inaccurate. This instead should have stated that these participants met the long stay definition. Information around their medical stability at the time of social work intervention was unknown, but we have assumed that these patients were medically fit for discharge as social workers in the usual model of care did not typically intervene until patients were experiencing delays to discharge.	Changes: Manuscript text under the subheading ‘study design, setting and participants’ has been amended to remove reference to control patients meeting pre-identified criteria for long stay and to add clarity around how controls were selected. New text has been highlighted in yellow.
2 Eligibility, outcome and bias		
Following on from the previous point, (a) how did the treatment group score on “pre-identified criteria determining their risk of long stay”? This is critical to the integrity of the study when LOS is the outcome of interest – if comparison group patients were identified on the basis of factors	Response: We acknowledge that the phrasing in the manuscript around pre-identified criteria determining risk of long stay was imprecise and has caused confusion. In addition to existing long stay patients, the treatment group consisted of patients deemed to be at risk of long stay under the new model of care. These ‘at risk’ patients were identified by ward social workers based on clinical opinion and informed by an audit of psychosocial barriers to	Changes: Further explanation has been added to the methods section under the ‘specialist social worker led model of care’ subheading. We have highlighted new text in yellow.

that predict LOS, and the treatment group differs systematically to the comparison group on these factors, then observed differences in outcome are more or less inevitable.	discharge. However, there was no systematic process underpinning this.	
For both groups, (a) clarify whether eligibility variables are baseline (hospital admission), pre-intervention (occur between admission and intervention) or outcome (e.g. LOS). (b) Defining the sample by outcomes such as LOS is problematic in terms of bias and policy relevance (https://www.ncbi.nlm.nih.gov/pmc/articles/PMC5034210/).	Response: Our aim was to create two comparable groups of patients. Prior to the intervention there was no process for the early identification of patients at risk of long stay with ward social workers typically becoming involved once patients had met the definition of long stay and were experiencing barriers to discharge. Following the introduction of the new model of care, a focus on early identification meant that patients could be identified, tracked and monitored prior to them meeting the long stay definition. We appreciate that the different eligibility variables may introduce bias and agree that this is a limitation of our study design. Unfortunately, as this study was conducted retrospectively, we were not able to establish consistent eligibility criteria in the pre/post intervention cohorts.	Changes: We have clarified eligibility variables for each group in the methods section, with new text highlighted in yellow under the subheading 'study design, setting and participants'. As per the response above, we have also included further detail around how intervention patients were put into Pathfinder under the subheading 'establishing a profile of long stay patients'. In the discussion section we have acknowledged that the selection of the control cohort is a key limitation of the study and discuss how this could have been improved upon with a prospective pre/poststudy design.
Consider presenting both groups according to all "pre-determined criteria" of long-stay risk in Table 1.	Response: This was not possible due to the absence of a systematic process to identify pre-determined criteria of long stay (see response to 2(a) above for further explanation.	
3 Timing bias		
The paper is strengthened by interest in timing-related bias but in reading the Wolkewitz reference (a) I struggled to follow how your study is relevant to "length bias". This seems to be a particular concern when there is a "Time-dependent entry criterion", i.e. an additional event has to occur between enrolment and exposure, e.g. being put on a ventilator where ventilator-related	Response: We have now moved to a simpler model of admission to discharge, with an intermediate state of "social worker intervention". We assume the time-dependent entry will be similar for both groups because they would be identified as at risk patients in similar ways, i.e., concerns of staff around barriers to discharge for those patients who were not quickly discharged.	Changes: Added a new figure that describes the three states.

infection is the exposure of interest (or Oscar nomination in the Oscar example).		
(a) I don't understand what your time-dependent entry criterion is, unless it's some kind of flag "this person is now at risk of long stay" that indicates social workers need but does not automatically trigger social worker involvement; if this is the case it is not sufficiently explained. (b) If adjusting for "length bias", consider an additional supplementary table where you follow Wolkewitz et al's schematic Table 2 (page 1177) and explain for each field how your study corresponds to the Oscar example.	Response: The time-dependent entry was too difficult to work out retrospectively. This could have been accurately collected in a prospective study. As stated above, we assume this entry time was similar for both groups.	Changes: Used a multistate model to estimate the additional length of stay.
(a) You don't mention Wolkewitz's other concern "time-dependent exposure bias" but look again at this. As written your paper gives no indication of difference in time-to-contact with social worker across the two groups. But if this differs there are a number of interpretations. On the one hand, if time-to-intervention is different between groups then this risks other biases whereby one group were in hospital longer (and so nearer to discharge, ceteris paribus) irrespective of the intervention. On the other hand, intervention timing has been shown	Response: We have assumed that the care was similar for both groups (in terms of both timing and intensity), except for the additional social worker intervention in one group. In the intervention group, the timing of the social worker intervention did vary between patients, and we have now used that data in the multistate model.	Changes: Used a multistate model to estimate the additional length of stay.

to be key to impact of consultative hospital interventions on utilisation (https://www.ncbi.nlm.nih.gov/pmc/articles/PMC5034210/), and if the time-to-contact is less in the treatment group this could itself be an indicator of intervention efficiency (assuming the right people are being identified).		
(a) Consider presenting both groups' time-to-contact with social worker in Table 1: (b) "The timing of all events should be recorded. Careful analysis and reporting of important details of the statistical analysis (including time origin and entry time) are required in the presence of time dependent study entries and exposures." (c) [What is new? Page 1172	Response: We have added summary statistics on the time between admission and involvement of the social worker.	Changes: Added a sentence to the results with the median and inter-quartile range of times between admission and involvement of the social worker.
4 Cohort matching and treatment effect estimation		
Concerns about eligibility criteria are compounded by large differences in characteristics of the treatment and comparison groups. At a bare minimum this requires more discussion in the limitations – do the authors think this variation occurred by chance or reflects systematic differences?	Response: We have now removed the surgery patients, so both groups were from "internal medicine".	Changes: Changed the analysis to exclude the surgery patients.
As well as the big age difference, the comparison group come from internal medicine and surgery wards, but treatment group came almost exclusively from internal medicine wards (Table 1). As well as an obvious collinearity risk, this begs the question: if your treatment group are (almost) all from the internal medicine	Response: See previous.	

wards, why not make internal medicine ward location an eligibility criterion for the comparison group? You'd ditch unrepresentative controls while substantively preserving your treatment group. The status quo invites heterogeneity between groups that is hard to justify (or, at least, is not justified currently).		
Also you seem to have included 'missing ward data' as a predictor in regression (Table 2). This requires more justification, especially as it's only for one subject. Are you controlling for some unobserved factor associated with this missingness? What happens if you drop this?	Response: Thank you for your observation. We have gone back to the records and located the 'missing ward data'	Change: Results tables and text have been revised accordingly
Both age and ward have a significant association with LOS in the primary analysis (Table 2). This seems a significant concern worth flagging in the limitations given that the groups differ on these factors. While these covariates were controlled for in multivariate regression the authors should consider using a matching technique such as propensity score weights to minimise confounding worries and isolate the estimated effect of the intervention (J Pain Symptom Manage. 2014 Oct;48(4):711-8).	Response: We have now used propensity score matching on age and gender	Change: Results table and text have been revised accordingly
5 Statistical analysis		
Primary analysis is summarised page 7, lines 43-52. Not clear why a survival analysis is used for a utilization variable. Utilization data are statistically awkward but log-transformation is not a particularly good way to address this (J Health Econ. 1998 Jun;17(3):283-95).	Response: Thanks for this interesting paper. We have now used a bootstrap model as suggested without transforming the dependent variable to estimate the difference in costs.	Change: Methods and results have been revised to reflect this change
Why not use nonlinear models, bootstrap the standard errors and report the estimated effect in days?		
I am also confused by the use of percentage reductions instead of day reductions.	Response: We are now using the absolute scale so this is no longer an issue.	

This appears a way to overcome the retransformation problem (J Health Econ. 1998 Jun;17(3):283-95) but I don't see that it does. A percentage is a ratio, but the ratio of ln(x) to ln(y) is not the same as the ratio of x to y so what is the advantage of using %? In what sense are % differences calculated on the log scale relevant?		
If % reductions are retained, justify this choice in more detail with respect to the retransformation problem and clarify in reporting if these are % of total LOS or % of LOS from the point of social worker interaction per "length bias" adjustment.	Response: We are now using the absolute scale so this is no longer an issue. Changes:	
Secondary analysis (later presented in Table 3) is not adequately covered pages 7-8. Explain both the aim and the approach in more detail; currently "Monte Carlo" appears for the first time in Results. See also next comment.	Response: We thank the reviewer for this comment and agree with this assessment.	Changes: We have included more detail to explain both the aim and approach in performing the secondary analysis as presented in Table 3. New text is highlighted in yellow in the last two paragraphs of the Methods section.
6 Table 3		
I got lost around Table 3. The results are summarised as "The cost to run the model of care for 12 months was \$228,876. Based on the modelled simulations, a total of 7,165 bed days were released over 12 months at an average cost of \$78 per bed day released (Table 3)." My reading of this is that if 7,165 bed days were saved at a cost of \$228,876 then the cost to release a bed day is $(228876/7165) = \\$32$. You need to spell out how you reached the \$78 figure because it is not intuitive from the results given. Per previous comment, better explanation of methods might help here.	Response: We thank the reviewer for highlighting this inconsistency which arose due to an error in reporting between different versions of the Monte Carlo simulation analysis. The reviewer is correct in calculating the cost per bed day released as \$32. Applying the same calculations, but adopting the revised length of stay figures (see explanation in response to major comment 5 above), this figure is now \$23.	Changes: This estimate has been revised in Table 3 and the accompanying text.
Also, how are estimated bed days calculated? You explicitly stay away from days in the primary analysis (Table 2), reporting instead % reductions. If calculating	Response: Bed days were previously calculated by applying the relative risk to the baseline length of stay in a Monte Carlo simulation. The model randomly sampled from the normally	Changes: These estimates have been revised in Table 3 and the accompanying text

days from log-transformed output is a problem in Table 2, how is it not in Table 3?	distributed log relative risk, and then took the exponent of this estimate to convert it back to a standard relative risk. Given the changes to the length of stay calculations as reported in Table 2 (detail in response to major comment 5 above) we have now applied the bootstrap length of stay estimates in calculating Table 3 outcomes.	
The introduction of the \$216 here is in any case too abrupt and inappropriate. If comparing cost per bed day saved against this \$216 benchmark was an aim at the outset, mention this in both the Introduction and the Methods. If it occurred only ex post then leave it for the Discussion and delete lines 13-21 on page 9.	Response: We agree with the reviewer's assessment. It was our original aim to apply a willingness to pay benchmark to the cost per bed day released, this has now been reflected in the Abstract, Introduction and Methods.	Changes: See changes reflected in the Abstract, Introduction and Methods (new text highlighted in yellow)
MINOR COMMENTS		
1 'Economic evaluation'		
The title of this article identifies it as an economic evaluation and this phrase recurs in framing the work conducted. Economic evaluation typically involves the assessment of a treatment's impact on costs (and often but not necessarily outcomes). The primary aim of this study is assessing effect on hospital LOS, and then this is reported in % terms. The authors concede this is not a CEA, since no outcomes are included, but an economic evaluation without outcomes is a *cost*-minimisation analysis. This primary analysis includes no costs. The authors add that "length of stay is now a well-recognised measure of resource use and efficiency in hospitals" (page 10, line 33). This may justify the study question but it does not make it an economic evaluation. Since LOS already appears in the title, consider changing "economic evaluation" to "cohort study" and removing	Response: We thank the reviewer for this comment but respectfully disagree that the term 'economic evaluation' is inaccurate. While we have not attributed costs to outcome measures, we have estimated costs associated with the social worker intervention. On reflection, we acknowledge that the term 'cost consequence analysis' (CCA) would provide a more precise description of the specific study design we have adopted. CCA is a type of economic evaluation where both the costs of an intervention and its non-monetised outcomes are separately accounted for and presented, leaving decision makers to come to their own judgements about their relative value and importance.	Changes: We have now changed the title and several references to the study design throughout from 'economic evaluation' to 'cost consequence analysis'. We have left in the references to this study being the first economic evaluation of a social work initiative as we feel this provides a more accurate description of the broader context in which this study sits.

all mention of economic evaluation from the text.		
2 Social worker involvement		
The manuscript provides good detail on the social worker model once patients are under social worker auspices and the supplementary file expands upon this. But I didn't understand how the social worker comes to be involved in treating a patient. Are they identifying suitable patients unilaterally via hospital databases and approaching teams with primary responsibility for patient care? Or are they involved at the invitation of that team? Or something else? And did this differ in the new model to the old? This is particularly important given that being seen by the social worker seems a prerequisite for membership of treatment and comparison groups (MAJOR COMMENTS 1 & 2).	Response: Referrals come from ward social workers via clinical consultations with the specialist social worker. These referrals were subsequently entered on Pathfinder. The specialist social worker also regularly collated a list of long stay patients from hospital data bases and then consulted with the different social work teams to identify those patients facing psychosocial barriers to discharge. In contrast to the old model of care, the new model was proactive rather than reactive and adopted a systematic approach to identifying long stay patients or those at risk of long stay.	Changes: This additional information has been added to the Methods section, highlighted in yellow under the subheadings 'Usual Model of Care (Control)' and 'Specialist Social Worker-led (New) Model of Care (Intervention)'
3 Who are the population to target?		
Fundamentally, the purpose of a study such as this is to answer the question 'should we give intervention I to population P?' The inadequate description of the population is a concern not only for rigour (MAJOR COMMENTS 1 & 2) but also usefulness to decision-makers. By the end of the study, the suggestion is that this model of hospital social work should be pursued but we have little information on who are the people this model will help. Only hospitals with such patients should employ this model, presumably, so we need to know more about who they are and how we identify them.	Response: We thank the reviewer for this feedback.	Changes: We have included some additional text in the methods section (as per responses to Major Comments 1 and 2 above) and also added two additional supplementary files.
4 How does it work?		
Some interpretation would also be welcome. The discussion focuses on contextual justification and	Response: We thank the reviewer for this suggestion	Changes: We have replaced the second paragraph of the discussion with some explanation around why we

resource concerns, which are all valid. But why do the authors think they see the association that they do? Consider adding some information on how better social work provision expedites hospital discharge for this particular population (while avoiding causal language). Also, consider commenting on how to identify these “at risk of long stay” patients to minimise their unnecessary stays. This material could replace the paragraph beginning line 46 page 9, which essentially restates why conducting the study is important; that paragraph could be merged into the introduction or excluded.		believe the intervention had an association with LOS, and how clinicians were able to identify patients at risk of long stay earlier in the admission.
5 Miscellanea		
-Time period is described as three months May to August (line 24, page 6). Give specific dates or this is a four-month period.	Response: We thank the reviewer for picking up this oversight.	Changes: Text has been changed accordingly on page 6 to June to August
-Sample size calculation is described as “86% power to detect a 40% decrease in length of stay assuming a two-sided 5% type 1 error” (page 6, line 34). If you are powering for detecting change only in a specific direction, isn’t this a one-sided test?	Response: It was a two-sided power calculation, but we have removed this sentence. The calculation was there to give an approximate idea of the number of required patients and the sample size was mostly based on the practical considerations of the number of patients seen by the new model of care, and the number of recent historical controls available.	
-Recruited sample (n=53) is mentioned in Methods (line 44, page 7). This should not be revealed until the results.	Response: We thank the reviewer for picking this up.	Changes: Sample size numbers have been removed from the Methods section.
-Sensitivity analyses. Were there any? Usually a good idea in utilization studies, especially with small samples. For example, given that the treatment group came almost exclusively from general medicine wards (Table 1), what happens if surgery patients are dropped from analysis?	Response: As suggested above, we have now removed the patients from the surgical ward.	
Reviewer 2 comments	Response	Changes to manuscript

This is an excellent topic for research. My main objection is with the methodology. The authors have used a historical control and they discuss this. They have 60 patients of which 59 are medical and 1 surgical vs all medical in the study group. It would strengthen the study significantly if only medical patients were used and the stats redone.	Response: As suggested, we have revised the analysis to include medical patients only.	Changes: Methods and results have been updated throughout the manuscript accordingly.
---	--	---

VERSION 2 – REVIEW

REVIEWER	Peter May Trinity College Dublin, Ireland
REVIEW RETURNED	23-Aug-2018

GENERAL COMMENTS	Thank you for the opportunity to review this resubmission of a hospital social worker model. The intervention is an interesting one, and the authors have made substantive efforts to address comments raised in the previous round. The biggest issue outstanding is that of eligibility and bias. The authors now acknowledge this risk (page 11 of 39, line 8+) but there appears little they can do about it: “Control patients were selected after having met the definition of long stay, while intervention patients were able to be selected without first meeting this definition.” Unfortunately I think this is a critical flaw. All comparison group patients had a long hospital stay, by definition. They are then compared *for length of stay* to an intervention group defined by other criteria. This introduces a systematic bias and renders the comparison of very low relevance. Lack of baseline data on the two groups further limits understanding of how they differ. OTHER COMMENTS 1 Title is inappropriate A cost-consequence analysis is one that examines intervention effect on costs and (some form of) patient outcome e.g. HRQOL. This study does not qualify. Moreover, ‘impact’ implies a causal study which this is not. Suggested alternative: Cohort study of a specialist social worker intervention on hospital utilization for patients at risk of long stay
---

	2 Recurring use of inappropriate terms: 'impact' and 'economic evaluation' This is not a RCT. Avoid causal language, e.g "we found that that a specialist social worker-led model of care reduced length of stay in an acute hospital setting." (page 11 of 39) Stick to "associations" not "impacts". This paper is not an economic evaluation. Primary analysis includes neither costs nor outcomes. These recur through the abstract and paper, misrepresenting the strength of analyses and evidence. 3 Miscellanea "older people's mobility, muscle strength and aerobic capacity can be adversely affected by just ten days of bed rest, which, alarmingly, translates into a loss of ten years of life." Extraordinary statement. It reads like 10 days of bed rest reduces life expectancy by ten years. Can this be true? More likely it reduces strength and aerobic capacity to that of someone 10 years older? Neither of these interpretations is explicitly supported by the abstract for the citation. Review and revise if necessary.
--	--

REVIEWER	Riyaz Jinnah Wake Forest University
REVIEW RETURNED	20-Aug-2018

GENERAL COMMENTS	This study presentation has been cleaned up nicely and now presents a succinct aim and conclusion.
--

VERSION 2 – AUTHOR RESPONSE

Reviewer 1 comments

The biggest issue outstanding is that of eligibility and bias. The authors now acknowledge this risk (page 11 of 39, line 8+) but there appears little they can do about it: "Control patients were selected after having met the definition of long stay, while intervention patients were able to be selected without first meeting this definition."

Unfortunately I think this is a critical flaw. All comparison group patients had a long hospital stay, by definition. They are then compared *for length of stay* to an intervention group defined by other criteria. This introduces a systematic bias and renders the comparison of very low relevance. Lack of

baseline data on the two groups further limits understanding of how they differ.

We thank the reviewer for their comment and agree that our selection of the historic control cohort is a key limitation of our study design that may introduce bias. Nonetheless, we believe that our paper provides an important contribution to the literature. It is the first study to provide quantitative evidence of the economic value of a hospital based social worker intervention. We have now acknowledged the limitations of our design in the conclusions section of abstract, cautioning the interpretation of our results and highlighting the need for further research to support our findings. We have also provided additional detail around the nature of this limitation in the 'Strengths and Limitations' section for further transparency. The discussion (pg 11, para 1) expands upon this limitation and highlights how the study may have adopted a stronger design; this may be of relevance to others considering how to conduct a similar study.

OTHER COMMENTS

1 Title is inappropriate

A cost-consequence analysis is one that examines intervention effect on costs and (some form of) patient outcome e.g. HRQOL. This study does not qualify. Moreover, 'impact' implies a causal study which this is not.

Suggested alternative:

Cohort study of a specialist social worker intervention on hospital utilization for patients at risk of long stay

We thank the reviewer for this comment and have adopted the alternative title as suggested.

2 Recurring use of inappropriate terms: 'impact' and 'economic evaluation'

This is not a RCT. Avoid causal language, e.g. "we found that that a specialist social worker-led model of care reduced length of stay in an acute hospital setting." (page 11 of 39) Stick to "associations" not "impacts".

This paper is not an economic evaluation. Primary analysis includes neither costs nor outcomes.

These recur through the abstract and paper, misrepresenting the strength of analyses and evidence.

We have amended the text to replace all references to the 'impact' of the intervention with 'association':

- Abstract: objective and conclusions
- Methods section: first para under 'Study design, setting and participants' subheading

We have removed all references to 'economic evaluation' and where appropriate replaced this with text that clarifies that we took an 'economic perspective':

- Strengths and limitations section: first point
- Introduction: first line of last paragraph
- Discussion: first sentence

3 Miscellanea

"older people's mobility, muscle strength and aerobic capacity can be adversely affected by just ten days of bed rest, which, alarmingly, translates into a loss of ten years of life."

Extraordinary statement. It reads like 10 days of bed rest reduces life expectancy by ten years. Can this be true? More likely it reduces strength and aerobic capacity to that of someone 10 years older?

Neither of these interpretations is explicitly supported by the abstract for the citation. Review and revise if necessary.

We have gone back to the cited paper and agree that we could improve upon our representation of its findings. We have now revised this sentence to read:

"In addition, older people's mobility, muscle strength and aerobic capacity can be adversely affected by just ten days of bed rest, which, alarmingly, translates into almost ten years of functional decline"

Editorial comments

We appreciate that reviewer 1's comments have remained quite negative; however, in light of the positive comments from the other reviewer, we felt we should give you the opportunity to respond to the criticisms and revise your manuscript appropriately. We ask that you state the limitation of the historical control group up front in the 'Conclusions' section of the abstract (along with it being in the 'Strengths and limitations' section).

We are grateful for the opportunity to respond to Reviewer 1's comments. As highlighted in our first response above, we agree with the reviewer's assessment around the limitation of our process for

selecting historical controls, this was unfortunately unavoidable and we have provided additional transparency around this limitation in the abstract and 'strengths and limitations' sections of the manuscript. Given this study provides the first reported estimates of the economic value of a hospital based social worker intervention we believe our findings provide an important contribution to the literature in this field, despite this limitation.

As requested, we have included additional text in the conclusions section of the abstract to explicitly acknowledge this limitation and call for further research to confirm our findings, as below:

"The limitations of our historic control cohort selection means that results should be interpreted with caution. Further research is needed to confirm these findings."